# Torquetenovirus in saliva: A potential biomarker for SARS-CoV-2 infection?

**Maria C. Mendes-Correa**[1☉], **Tania Regina Tozetto-Mendoza**[1☉*], **Wilton S. Freire**[1], **Heuder G. O. Paiao**[1], **Andrea B. C. Ferraz**[2], **Ana C. Mamana**[1], **Noely E. Ferreira**[1], **Anderson V. de Paula**[1], **Alvina C. Felix**[1], **Camila M. Romano**[1], **Paulo H. Braz-Silva**[1,3], **Fabio E. Leal**[2], **Regina M. Z. Grespan**[2], **Ester C. Sabino**[4], **Silvia F. Costa**[5], **Steven S. Witkin**[1,6]

**1** Faculdade de Medicina da Universidade de Sao Paulo-Laboratório de Investigacao Medica em Virologia (LIM52)-Instituto de Medicina Tropical de Sao Paulo, Universidade de São Paulo, São Paulo, Brazil,
**2** Faculdade de Medicina da Universidade Municipal de Sao Caetano do Sul, São Paulo, Brazil, **3** Faculdade de Odontologia da Universidade de São Paulo – Departamento de Estomatologia, São Paulo, Brazil,
**4** Faculdade de Medicina da Universidade de São Paulo-Laboratório de Investigação Médica em Virologia (LIM46)-Instituto de Medicina Tropical de Sao Paulo, São Paulo, Brazil, **5** Faculdade de Medicina da Universidade de Sao Paulo-Laboratório de Investigação Médica em Virologia (LIM49)-Instituto de Medicina Tropical de Sao Paulo, São Paulo, Brazil, **6** Department of Obstetrics and Gynecology, Weill Cornell Medicine, New York, New York, United States of America

☉ These authors contributed equally to this work.
\* tozetto@usp.br

**Data Availability Statement:** All relevant data are within the manuscript.

**Funding:** The work was supported by FAPESP 2020/05623 (Fundação de Amparo à Pesquisa do

## Abstract

Torquetenovirus (TTV) is present in biological fluids from healthy individuals and measurement of its titer is used to assess immune status in individuals with chronic infections and after transplants. We assessed if the titer of TTV in saliva varied with the presence of SARS-CoV-2 in the nasopharynx and could be a marker of COVID-19 status. Saliva from 91 individuals positive for SARS-CoV-2 in nasal-oropharyngeal samples, and from 126 individuals who were SARS-CoV-2-negative, all with mild respiratory symptoms, were analyzed. Both groups were similar in age, gender, symptom duration and time after symptom initiation when saliva was collected. Titers of TTV and SARS-CoV-2 were assessed by gene amplification. Loss of smell (p = 0.0001) and fever (p = 0.0186) were more prevalent in SARS-CoV-2-positive individuals, while sore throat (p = 0.0001), fatigue (p = 0.0037) and diarrhea (p = 0.0475) were more frequent in the SARS-CoV-2 negative group. The saliva TTV and nasal-oropharyngeal SARS-CoV-2 titers were correlated (p = 0.0085). The TTV level decreased as symptoms resolved in the SARS-CoV-2 infected group (p = 0.0285) but remained unchanged in the SARS-CoV-2 negative controls. In SARS-CoV-2 positive subjects who provided 2–4 saliva samples and in which TTV was initially present, the TTV titer always decreased over time as symptoms resolved. We propose that sequential TTV measurement in saliva is potentially useful to assess the likelihood of symptom resolution in SARS-CoV-2-positive individuals and to predict prognosis.

Estado de São Paulo) and Virology laboratory (LIM 52) of the Tropical Medicine Institute, São Paulo University, Medicine School, Brazil.

**Competing interests:** The authors have declared that no competing interests exist.

## Introduction

Torquetenovirus (TTV) is a small, single stranded DNA virus present in multiple body fluids of apparently healthy individuals [1, 2]. It is generally considered to be a non-pathogenic endogenous virus and is not associated with any known pathology. However, since its titer in body fluids varies with the local immune status, determination of the TTV titer has been shown to be a sensitive quantitative indicator of the likelihood of rejection following organ transplantations [3–5]. The higher the TTV titer the greater is the level of immune suppression. TTV measurement also has value outside of organ transplantation. Its detection and titer have been utilized as a marker of immune function in chronic diseases such as liver cirrhosis [6] or HIV [7].

TTV has been shown to be present in human saliva [8, 9]. In a patient who underwent an allogeneic hematopoietic stem cell transplant, detection of TTV was more frequent and the TTV titer was higher in saliva as opposed to plasma [9]. This led the authors to suggest that quantitation of TTV in saliva may be of value in predicting immunocompetence after engraftment [8].

The worldwide pandemic of Coronavirus disease 2019 (COVID-19) is due to infection by severe acute respiratory syndrome virus-2 (SARS-CoV-2) [10]. To our knowledge there is only a single study, a case report, on the association between the TTV titer and SARS-CoV-2 [11]. In a 42 year old male kidney transplant patient who exhibited only mild symptoms of COVID-19, his peripheral blood TTV titer was highest when SARS-CoV-2 was first detected in a nasopharyngeal swab. The TTV titer was greatly reduced when symptoms resolved and his nasopharyngeal swab became SARS-CoV-2- negative. The patient remained on immunosuppressive medication both at the onset and after resolved COVID-19. When the SARS-CoV-2 infection was resolved the extent of lymphopenia decreased and TTV proliferation was reduced. SARS-CoV-2 has been shown to contribute to lymphopenia [12, 13]. These observations suggest that measurement of the TTV titer may have value as a surrogate marker of SARS-CoV-2 infection.

In the present study we measured the TTV titer over time in saliva from mildly symptomatic individuals who were positive for SARS-CoV-2 in nasal-oropharyngeal samples. Controls were individuals with similar symptoms who were negative for SARS-CoV-2. We hypothesized that the salivary TTV titer in the SARS-CoV-2 patients would vary during the course of their disease and, thus, might be of value as a measurement of COVID-19 progression or resolution.

## Materials and methods

### Ethics, study design, setting and population

The study was approved by the local ethics committee: Comissão Nacional de Ética em Pesquisa do Ministério da Saúde do Brasil (CONEP), protocol No. CAAE 30419320.7.0000.0068, dated April, 18, 2020) and all subjects provided written informed consent. This observational, prospective study used a convenience samples from participants in *The Corona São Caetano Program*, a primary care initiative offering COVID-19 care to all residents of São Caetano do Sul, Brazil [14]. Briefly, residents with symptoms consistent with COVID-19 were encouraged to contact the Corona São Caetano platform via a website or by phone. Subjects were excluded from study if they presented symptoms related to an allergy or bacterial infection and/or used antibiotics or other medications in the two weeks prior to sample collection. All subjects from study have no underlying condition. The respondents were invited to complete an initial screening questionnaire that included information on type, onset and duration of symptoms. Those with symptoms consistent with COVID-19 were contacted by a medical student for

further risk assessment. Individuals meeting pre-defined criteria for mild COVID-19 were offered a home visit in which a self-collected nasal-oropharyngeal swabs and a saliva sample were obtained for analysis. Samples were collected from May 5 to May 30, 2020.

## Sample collection

Nasal-oropharyngeal samples were collected from both nostrils and the oropharynx using commercial flocked swabs with plastic applicators (Goodwood Medical Care Ltd., Jinzhou, China). Saliva samples were collected using a cotton pad device–Salivette™ (Sarstedt AG & CO. KG, Nümbrecht, Germany). To provide guidance on self-collection procedures, a link to an instructional video was sent to each participant before the home visit. Briefly, subjects were instructed to insert the swabs into both nostrils and the posterior region of the mouth and put all swabs into a tube containing a saline solution. For saliva collection, they were instructed to vigorously chew a cotton pad for one minute before its placement into a separate tube. Participants were instructed to avoid eating, drinking or toothbrushing at least one hour before the saliva collection. In accordance with the Corona São Caetano Program procedures, all samples were collected in the morning, immediately placed in a cool box (2–8˚C) and stored at 4˚C in a refrigerator until shipment to the lab that same afternoon by a specialized carrier. The date that the swabs were collected was noted. Several individuals positive for SARS-CoV-2 in their saliva were requested to collect additional saliva samples over a 30 day period, while those who tested negative were followed up in the primary health-care program. The patients were asked to contact the platform for a new consultation if they developed new symptoms.

## Analysis for SARS-CoV-2

Nasal-oropharyngeal samples were handled according to laboratory biosafety guidelines. The specimens underwent RNA extraction by using the EasyMag® automatized extractor (Nucli-SENS® easyMag® bioMérieux, Durham, NC), according to the manufacturer's instructions. All samples were deemed suitable for amplification by RT-PCR based on analysis of the internal control consisting of primers and probe for the human Ribonuclease P gene (GenBank accession number NM_006413), as described [15]. Samples were then subjected to RT-PCR (RealStar® SARS-CoV-2 RT-PCR Kit 1.0, Altona Diagnostics) in order to provide a qualitative diagnosis for SARS-CoV2. For in house absolute quantification by real-time polymerase chain reaction (qPCR), specific primers and probes for the E-gene and NP-gene of SARS-CoV2 were synthesized, as described previously [16, 17]. HPLC-purified oligonucleotides sequences of both genes of SARS-CoV-2 were synthesized as follows: E-gene -synthetic oligo (E): 5´TTCG TATATTGCAGCAGTACGCACACCGTATCGAAGCGCAGTAAGGATGGCTAGTGTATGCGTACG CTATTAACTATTAACGTACCTGTCTGATA -3' and NP-gene -synthetic oligo (HKU_NP): 5'- TTCGTCGAAGGTGTGACTTCCATGCGTATCCGCAAATTGCACAATTTGCATGCGTAAT CAGTTCCTTGTCTGATTACTGATA–3'.

Standard curves with known amounts of the synthetic sequences were generated for qPCR by a methodology previously described [18, 19]. The in house assay was performed by using TaqMan™ Fast one virus master mix protocol (Thermo Fisher Scientific®, Austin, USA). The data were analyzed using QuantStudio Design & Analysis Software v.1.4.1. The range of detection of the E-gene was 3.5 to 10.1 $\log_{10}$ copies per mL, with an amplification efficiency value of 91.9% (slope of -3.5, y-intercept of 40.82, R2 of 0.99). For the NP-gene, the range of detection was 4.7 to 11.7 $\log_{10}$ copies per ml, with an amplification efficiency value of 98.0% (slope of -3.37, y-intercept of 39.65, R2 of 0.95). The lower limit of detection of the qPCR assay was 3,049 copies per ml (3.5 $\log_{10}$ copies per ml). The viral load data were based on the E-gene findings. The NP-gene analysis was used as a confirmatory test to verify the viral load.

### TTV titer analysis

A cotton pad device–Salivette™ (Sarstedt AG & CO. KG, Nümbrecht, Germany) was used for saliva collection, according to the manufacturer's instructions. The total DNA from salivary fluid was extracted and purified by using the EasyMag automatized platform (NucliSENS® easyMag® bioMérieux, Durham, NC). All DNA samples were deemed suitable for DNA amplification by PCR based on analysis of the internal control. For TTV detection in saliva, TTV-specific probe and primers were performed as previously described by Maggi et al [20]. A standard curve with known amounts of a synthetic DNA was performed for TTV absolute quantification by real-time polymerase chain reaction as previously described [21]. The TTV DNA amplification was based on the TaqMan™ Universal PCR master mix protocol (Thermo Fisher Scientific, Warrington, UK). The data were analyzed using QuantStudio Design & Analysis Software v.1.4.1. The range of detection of TTV in saliva was 1.6 to 7.4 $\log_{10}$ copies per ml. Positive and negative controls for TTV in saliva were obtained from samples stored in the repository at the Virology Laboratory (Institute of the Tropical Medicine of the Medicine School of the São Paulo University, Brazil). The lower limit of detection of the qPCR assay for TTV was 40 copies per ml (1.6 $\log_{10}$ copies/ml).

### Statistics

Differences in discrete variables between individuals positive or negative for SARS-CoV-2 were analyzed by Fisher's exact test. Continuous variables were analyzed by the Mann-Whitney test. Differences in the TTV titer over time was analyzed by the Kruskal-Wallis test. Correlations between TTV and SARS-CoV-2 titers were analyzed by the Spearman rank correlation test. A p value <0.05 was considered significant.

## Results

The characteristics of the study population are shown in Table 1. There was a small increased prevalence of females in both the 91 SARS-CoV-2- positive patients (54.9%) and the 126 negative subjects (61.1%). Mean age was comparable (42.9 years and 41.4 years) in positive and negative participants, respectively. Similarly, the median time (range) from symptom onset to resolution was 4 (1–10) days in the SARS-CoV-2-positives and 4 (1–30) days in the controls. None of these differences were significant (p>0.05).

The frequency of specific symptoms manifested by individuals in both groups is described in Table 2. Most self-reported symptoms were comparable and independent of SARS-CoV-2 status. The prevalence of fever, designated as a temperature > 37.5˚C (p = 0.0186) and loss of smell (p = 0.0001) were higher in SARS-CoV-2-positive subjects while a sore throat (p = 0.0001), fatigue (p = 0.037) and diarrhea (p = 0.0475) were more common in the negative controls. Among the COVID-19 patients, loss of smell was more frequent in men (78.0%) than in women (52.9%) (p = 0.0138). Fever was also more common in the male COVID-19 patients (56.1% versus 38.1% in females), but this did not reach statistical significance (p = 0.0773).

**Table 1. Characteristics of the study population based on the SARS-CoV2 status.**

| Characteristic | | SARS-CoV-2 Positive (n = 91) | SARS-CoV-2 Negative (n = 126) | P value |
|---|---|---|---|---|
| Gender | Male No.samples (%) | 41 (45.1) | 49 (38.9) | p>0.05 |
| | Female No. samples (%) | 50 (54.9) | 77 (61.1) | p>0.05 |
| Age | Mean (Standard Desviation) | 42.9 (16.3) | 41.4 (14.7) | p>0.05 |
| Duration of symptoms | Median (range) | 4 (1–10) days | 4 (1–30) days | p>0.05 |

**Table 2. Specific symptoms in individuals in association with nasopharynx SARS-CoV-2 status.**

| Symptom | SARS-CoV-2 Positive (n = 91) | SARS-CoV-2 Negative (n = 126) | P value |
|---|---|---|---|
| Shortness of breath | 1 (1.1%) | 1 (0.7%) | p>0.05 |
| Fever | 49 (54.6%) | 47 (37.3%) | 0.0186 |
| Cough | 68 (74.7%) | 92 (73.0) | p>0.05 |
| Sore throat | 40 (44.0%) | 89 (70.6%) | 0.0001 |
| Nasal congestion | 58 (63.7%) | 72 (57.1%) | p>0.05 |
| Runny nose | 44 (48.4%) | 74 (58.7%) | p>0.05 |
| Headache | 68 (74.7%) | 98 (77.8%) | p>0.05 |
| Fatigue | 49 (53.8%) | 93 (73.8%) | 0.0037 |
| Weakness | 41 (45.1%) | 65 (51.6%) | p>0.05 |
| Anorexia | 39 (42.9%) | 59 (46.8%) | p>0.05 |
| Myalgia | 63 (69.2%) | 85 (67.5%) | p>0.05 |
| Arthralgia | 43 (47.3%) | 57 (45.2%) | p>0.05 |
| Diarrhea | 14 (15.4%) | 34 (27.0%) | 0.0475 |
| Nausea | 23 (25.3%) | 33 (26.2%) | p>0.05 |
| Vomiting | 5 (5.5%) | 10 (7.9%) | p>0.05 |
| Loss of smell | 68 (74.7%) | 46 (36.5%) | 0.0001 |
| Loss of taste | 38 (41.8%) | 38 (30.2%) | 0.0617 |

TTV was identified in saliva from 54.9% of SARS-CoV-2-positive subjects and in 62.7% of the negative controls. Variations in the saliva TTV titer as a function of the time from symptom onset, utilizing a single sample from each subject, is summarized in Table 3. In the first 9 days after symptom initiation the median $\log_{10}$ TTV titer/ml was relatively constant. However, starting on day 10 the TTV titer became greatly reduced in the SARS-CoV-2-positive subjects (p = 0.0285) but remained unchanged in the controls (p = 0.5255). In addition, the saliva TTV and nasal-oropharyngeal SARS-CoV-2 levels in the COVID-19-positive patients were correlated (p = 0.0085).

More than one saliva sample was available from 10 subjects. The decrease in TTV titer over time in these individuals is shown in Fig 1. In all cases, paralleling the findings in Table 3, there was a marked decrease in the TTV titer in individual patients as symptoms resolved and SARS-CoV-2 levels decreased.

## Discussion

TTV was detected in saliva from individuals who were positive for SARS-CoV-2 in their nasopharynx, and there was a significant decrease in the TTV titer as their respiratory symptoms resolved and their SARS-CoV-2 level decreased. Thus, in parallel to results of studies on TTV detection in transplant patients and in those with chronic infections, it is reasonable to propose

**Table 3. Torquetenovirus titer in saliva as a function of days from symptom initiation in relation to SARS-CoV-2 status.**

| Days | SARS-CoV-2 positive | | SARS-CoV-2 negative | |
|---|---|---|---|---|
| | No. Samples | Median $\log_{10}$ TTV titer/ml (range) | No. samples | Median $\log_{10}$ TTV titer/ml (range) |
| 3–5 | 27 | 3.3 (0, 6.1) | 54 | 3.6 (0, 7.4) |
| 6–7 | 19 | 2.0 (0, 6.2) | 26 | 4.4 (0, 7.3) |
| 8–9 | 16 | 3.8 (0, 5.7) | 20 | 3.6 (0, 7.4) |
| 10–16 | 20 | 0.0 (0, 5.2) | 21 | 3.9 (0, 6.2) |
| >16 | 8 | 0.0 (0, 2.5) | 5 | 5.1 (0, 5.8) |

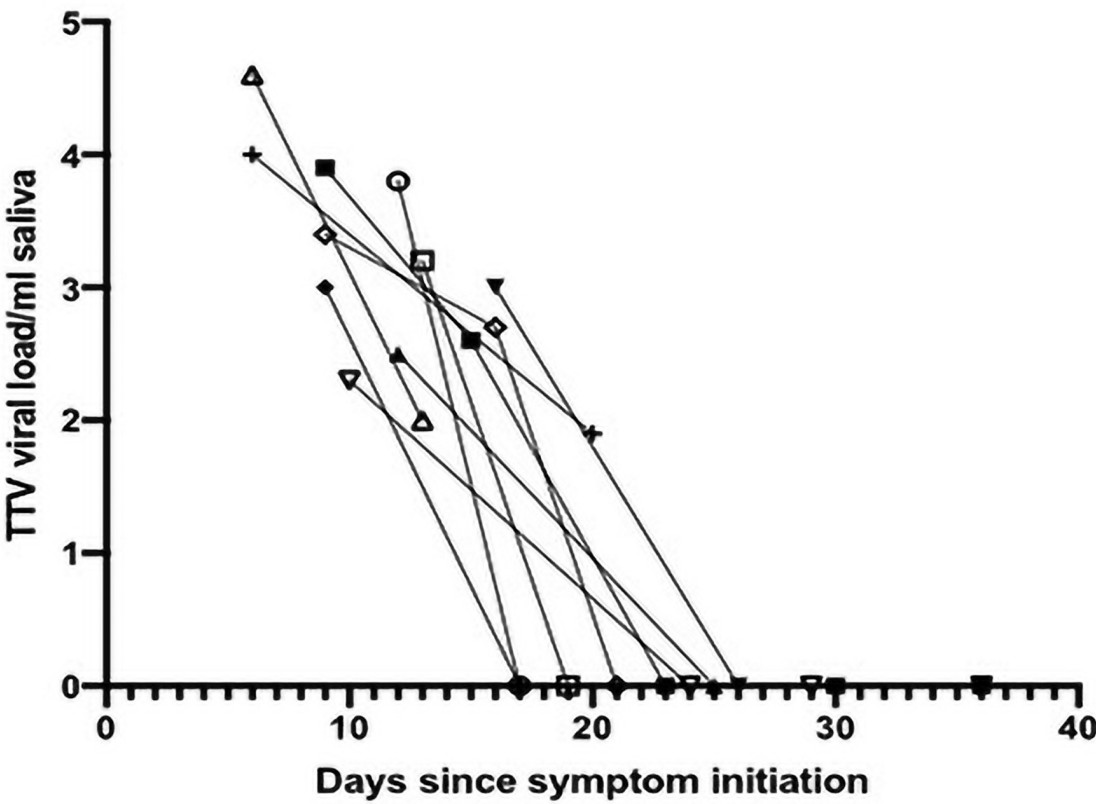

**Fig 1. Decrease in salivary TTV titer ($\log_{10}$ titer/ml) in individuals positive for SARS-CoV-2 as a function of time from symptom onset.** The TTV titer was determined in saliva samples that were obtained from 10 study subjects at more than one time point, and who were positive for TTV in their initial sample. The spaghetti plot details the change in titer for each subject at the times of saliva collection.

that the TTV titer in saliva is a surrogate biomarker for immune system activation in the nasopharynx. An initial high TTV titer at symptom initiation in SARS-CoV-2-infected patients is indicative of a localized SARS-CoV-2-mediated immune suppression, while a subsequent decrease in TTV parallel to the resolution of symptoms and the SARS-CoV-2 infection denotes an elevation in local immunity. The finding of a strong correlation between the salivary TTV titer and the nasopharyngeal SARS-CoV2 titer adds further support to this hypothesis.

In the present study it was difficult to differentiate between individuals who were positive or negative for SARS-CoV-2 on the basis of symptoms. A marked exception was a self-reported loss of smell. This was very highly correlated with detection of SARS-CoV-2, consistent with findings from other studies (reviewed in ref. [22]). The higher prevalence of a loss in smell in male COVID-19 patients is consistent with studies suggesting that symptoms of SARS-CoV-2 infection may be more severe in males [23].

Strengths of the present study include the collection of saliva and nasal-oropharyngeal samples from individuals with similar symptoms seen at a single center and in whom SARS-CoV-2 status was unknown. All samples were tested for TTV and SARS-CoV-2 by the same group of investigators who were blinded to all clinical data. The paired analysis of patients subsequently diagnosed with COVID-19 with a comparable non-infected control group provides added reliability to data interpretation. This study has several limitations. We do not have measurements of any immune mediators in the saliva samples. Therefore, our proposed mechanism for the decrease in TTV titer with symptom resolution in the COVID-19 patients, i.e., a

reduction in immune suppression, requires further verification. An alternate explanation of our findings is that the observed change in salivary TTV level was merely an indicator of the presence or resolution of a mild respiratory tract infection and was unrelated to SARS-CoV-2. Other viruses [24], including different coronavirus species [25], may also have been present in the nasopharynx of our study population. Thus, the possible influence of viral co-infections on SARS-CoV-2-related salivary TTV levels remains undetermined. We would stress, however, that the uniform responses in the SARS-CoV-2 positive and negative patients suggest that any influence of other co-infections is likely to be relatively minor. A parallel diminution in TTV titer over time after resolution of respiratory symptoms was not observed in the SARS-CoV-2 negative individuals. This strongly suggests that immune alterations associated with a mild respiratory tract infection are insufficient to have a noticeable influence on TTV levels in saliva. Lastly, despite the apparent consistency of the findings, the use of saliva samples from different subjects to create a timeline of TTV changes, rather than using samples from the same individuals, is not ideal.

In conclusion, we propose that measurement of the saliva TTV level over time in individuals positive for SARS-CoV-2 may be of value in ascertaining variations in local immune status and as predictors of prognosis.

## Acknowledgments

We thank the staff of the *Laboratório de Investigação Médica em Virologia* (LIM52)—Instituto de Medicina Tropical de Sao Paulo—Faculdade de Medicina da Universidade de São Paulo for technical support.

## Author Contributions

**Conceptualization:** Maria C. Mendes-Correa, Tania Regina Tozetto-Mendoza, Andrea B. C. Ferraz, Paulo H. Braz-Silva, Fabio E. Leal, Ester C. Sabino, Silvia F. Costa, Steven S. Witkin.

**Data curation:** Maria C. Mendes-Correa, Tania Regina Tozetto-Mendoza, Heuder G. O. Paiao, Ana C. Mamana, Noely E. Ferreira, Camila M. Romano, Steven S. Witkin.

**Formal analysis:** Maria C. Mendes-Correa, Tania Regina Tozetto-Mendoza, Wilton S. Freire, Heuder G. O. Paiao, Andrea B. C. Ferraz, Noely E. Ferreira, Anderson V. de Paula, Camila M. Romano, Steven S. Witkin.

**Funding acquisition:** Maria C. Mendes-Correa, Ester C. Sabino.

**Investigation:** Maria C. Mendes-Correa, Tania Regina Tozetto-Mendoza, Wilton S. Freire, Heuder G. O. Paiao, Andrea B. C. Ferraz, Ana C. Mamana, Noely E. Ferreira, Anderson V. de Paula, Alvina C. Felix, Fabio E. Leal, Regina M. Z. Grespan, Steven S. Witkin.

**Methodology:** Maria C. Mendes-Correa, Tania Regina Tozetto-Mendoza, Wilton S. Freire, Heuder G. O. Paiao, Andrea B. C. Ferraz, Ana C. Mamana, Noely E. Ferreira, Anderson V. de Paula, Alvina C. Felix, Camila M. Romano, Paulo H. Braz-Silva, Steven S. Witkin.

**Project administration:** Maria C. Mendes-Correa, Tania Regina Tozetto-Mendoza, Ester C. Sabino, Silvia F. Costa, Steven S. Witkin.

**Resources:** Maria C. Mendes-Correa, Heuder G. O. Paiao, Andrea B. C. Ferraz, Ester C. Sabino, Steven S. Witkin.

**Software:** Andrea B. C. Ferraz.

**Supervision:** Maria C. Mendes-Correa, Tania Regina Tozetto-Mendoza, Fabio E. Leal, Regina M. Z. Grespan, Silvia F. Costa, Steven S. Witkin.

**Validation:** Maria C. Mendes-Correa, Tania Regina Tozetto-Mendoza, Wilton S. Freire, Heuder G. O. Paiao, Ana C. Mamana, Noely E. Ferreira, Paulo H. Braz-Silva, Steven S. Witkin.

**Visualization:** Steven S. Witkin.

**Writing – original draft:** Maria C. Mendes-Correa, Tania Regina Tozetto-Mendoza, Steven S. Witkin.

**Writing – review & editing:** Maria C. Mendes-Correa, Tania Regina Tozetto-Mendoza, Wilton S. Freire, Heuder G. O. Paiao, Andrea B. C. Ferraz, Ana C. Mamana, Noely E. Ferreira, Anderson V. de Paula, Alvina C. Felix, Camila M. Romano, Paulo H. Braz-Silva, Fabio E. Leal, Regina M. Z. Grespan, Ester C. Sabino, Silvia F. Costa, Steven S. Witkin.

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
