## [Decision Letter · Decision Letter 0]

13 Apr 2021

PONE-D-21-05788

Torquetenovirus in saliva: a biomarker for progression or resolution of SARS-CoV-2 infection?

PLOS ONE

Dear Dr. Tozetto Mendoza,

Thank you for submitting your manuscript to PLOS ONE. After careful consideration, we feel that it has merit but does not fully meet PLOS ONE’s publication criteria as it currently stands. Therefore, we invite you to submit a revised version of the manuscript that addresses the points raised during the review process.

I have received the reviews of your manuscript. While your paper addresses an interesting question, the reviewers stated several concerns about your study and did not recommend publication in its present form.  One reviewer has concern regarding the sample collection and control population, another reviewer also has concern about the matched control group.  Moreover, other reviewers have identified numerous issues where additional experimentation and documentation is needed. Please see the reviewers’ insightful comments for details and these comments need to be addressed carefully.  In addition, I would like the authors to address whether human subject research approval is needed for collecting the saliva samples? Also, the figure legend needs more detail in describing the symbols.  Additionally the quality of the language needs to be improved, there are quite a few awkward sentences throughout the manuscript.  Please have a fluent, preferably native, English-language speaker thoroughly copyedit your manuscript for language usage, spelling, and grammar.

We look forward to receiving your revised manuscript.

Kind regards,

Baochuan Lin, Ph.D.

Academic Editor

PLOS ONE

Journal Requirements:

"NO,

Reviewers' comments:

Reviewer's Responses to Questions

**Comments to the Author**

1. Is the manuscript technically sound, and do the data support the conclusions?

Reviewer #1: Partly

Reviewer #2: Partly

Reviewer #3: Yes

Reviewer #4: Partly

2. Has the statistical analysis been performed appropriately and rigorously? 

Reviewer #1: N/A

Reviewer #2: N/A

Reviewer #3: Yes

Reviewer #4: Yes

3. Have the authors made all data underlying the findings in their manuscript fully available?

Reviewer #1: No

Reviewer #2: No

Reviewer #3: Yes

Reviewer #4: Yes

4. Is the manuscript presented in an intelligible fashion and written in standard English?

Reviewer #1: Yes

Reviewer #2: Yes

Reviewer #3: Yes

Reviewer #4: Yes

5. Review Comments to the Author

Reviewer #1: Thanks for the opportunity to review the manuscript. The research idea was novel and attractive. However, the title is not completely compatible with the data and the methods that was provided in the manuscript. I have some comments and questions:

1- About the SARS-CoV-2 sampling, as I found the patients were instructed and did the sampling by themselves, using nasopharyngeal swabs. This method (nasopharyngeal swabs) is a sensitive sampling method but is painful and disgusting. How could you be sure that samples were collected correctly? I mean, how could you be sure that the negative samples were not false negative results?

2- We are in the pandemic of COVID-19, could we assume a symptomatic individual as a control just with a single SARS-CoV-2 negative PCR?

3- It was a good idea to select the controls among symptomatic individuals who had negative SARS-CoV-2 PCR, but we need more information about the cause of disease in controls because an allergic rhinitis would be different from common cold. Moreover, some of the controls could have other strains of corona virus. I afraid the controls are not homogenous.

4- In line 97, you wrote “Several individuals positive for SARS-CoV-2 in their saliva were requested to collect additional saliva samples over a 30-day period.”, what about individuals with negative PCR?

5- About the clinical sign and symptoms, did the patient measured his/her body temperature? If yes, is this a sensitive method to divide patients to febrile and afebrile?

6- Could we have more information about the drug history and background disease of both cases and controls? Because it might affect the TTV viral load.

7- In table 3, you showed that the SARS-CoV-2-positive patients had the highest median TTV viral load on day 8-9 while the viral load was lower in day 6-7, how could you interpret this finding considering the normal immunologic response to SARS-CoV-2 infection?

8- Could you describe, why you divided the study course to days 3-5, 6-7, 8-9, 10-16,…? Why the intervals are not equal?

9- You only provided data about the kinetics of SARS-CoV-2 virus, what about the clinical course of disease in different time points, data on hospital admission, or mortality?

10- About the figure 1, this is a spaghetti plot showing the kinetics of TTV, if each shape is representative for a single patient, I afraid you had a lot of missing data. Could you provide the figure legend with more details about the plot and shapes?

11- You might need to provide the immunologic markers, as well as more information about the clinical course of disease, if you aim to conclude that TTV is a marker of local immune status and as predictors of prognosis.

Reviewer #2: In this manuscript, Mendes-Correa et al. focuse on the attempt to demonstrate that TTV level in saliva from COVID-19 patients might be used as biomarker for progression or resolution of SARS CoV-2 infection. Although the topic may be of interest, the way it is treated is disappointing and a number of issues makes weak the manuscript. Overall, in the current form, the manuscript does not reach the quality standard needed for publication

Reviewer #3: Interesting study from a single center cohort about the association of TTV titer in saliva with SARS-CoV-2 status.

1) the major limitation is already discussed in the manuscript:

the SARS-CoV-2 negative group - all with "mild respiratory symptoms" was not further characterized.

Especially testing with a multiplex-PCR for other respiratory viruses would help to interpret the data in the manuscript, especially since TTV levels seem to stay unchanged in the SARS-CoV-negative group day 10-16 and >16 in contrast to the SARS-CoV-2 positive group.

If saliva samples are still available, I would recommend performing these analyses in both groups.

The manuscript could be further improved if a matched control group without respiratory symptoms would also be tested for TTV in saliva.

2) line 49: it is true that TTV has been described as an indicator of immune function mainly after organ transplantation, but there is also literature on TTV as a marker of immune function in chronic diseases such as liver cirrhosis or HIV. This could be briefly elaborated by the authors to show that looking at TTV levels could also have a value outside of organ transplantation.

3) line 53/54: should be probably ref. 6 (TTV in HSCT recip.); "predicting lymphocyte status" is somewhat obscure to me, would suggest immune function, immunocompetence or immunreconstitution.

4) line 191: it would be helpful to provide the number of subjects with more than one saliva sample, as this is rather small and therefore a cautious interpretation is required

5) line 203: would remove "anti-SARS-CoV-2" as no rationale is provided that TTV level reflects specific anti-SARS-CoV-2 immune activity - considering the literature, it is more likely a marker of overall immunocompetence

Reviewer #4: The authors describe the evolution of torque tenovirus viral load in saliva related to Sars-CoV 2 viral load in nasopharyngeal swabs, in an effort to find markers for immunosuppression. They also compare TTV levels at different times post initiation of symptoms for the covid 19 population (mild covid) and a control population with a mild respiratory disease.

Second, the authors show in a subpopulation of the covid 19 group that TTV viral load decreases with the disparition of symptoms. As a consequence, they consider that TTV viral load in saliva may be suitable to describe rhinopharyngeal local immune response. And they refer to studies comparing blood TTV viral load and evolution of immunity /rejection in transplant patients.

The idea is interesting, because simple tests to evaluate local immunity are very important to identify potential patients that could evolve toward high viral loads and covid worsening. The data concerning the evolution of TTV viral load in relation to SARS-CoV 2 viral load is convincing. This is, to my knowledge, the first paper describing TTV load in saliva for covid patients .

Although, the paper suffers from some weaknesses that could be overcome:

First : the authors did not justify the use of saliva instead of the Nasopharyngeal samples for TTV viral load.Although they refer to its use in previous comparisons with plasma levels in transplant patients to predict global immunity which is very different. They use the same extraction method . We understand of course that saliva is easier and potentially more reproducible for auto collection, but the TTV load values, in saliva, are highly variable in both populations and as NOP sampling was performed, they could have compared TTV load in both samples, to comfort their hypothesis.

Second : The large panel of viralloads at each time, coming from the fact that there was only one sample per patient and thus at each time we compare different patients make the group comparison less significant, the authors should comment on this and provide a graph in addition to the table to show the repartition of viral loads within time in both groups.

Third : The lowering of TTV viral load from sequential samples in the covid patients cannot be compared with that of the non-covid group. In addition we do onot have the diagnosis (VRS may be different from rhinovirus for example in terms of local immunity, samely pharyngitis or thracheitis may be different from coryza?) Theu could add the diagnosis if available.

Fourth : The authors have choosen a mild covid group for a better comparison with mild respiratory disease. Could they correlate the initial TTV load or the maximum TTV load , with the duration of symptoms , or the delay to SARS-CoV 2 undetectability? This could be very interesting to predict persistent infections.

Minor comments : some text errors or form errors are present in the text and bibliography, and a cautious review is necessary before submission.

6. PLOS authors have the option to publish the peer review history of their article (what does this mean?). If published, this will include your full peer review and any attached files.

Reviewer #1: **Yes: **Omid Rezahosseini

Reviewer #2: No

Reviewer #3: No

Reviewer #4: No

---

## [Author Response · Author response to Decision Letter 0]

7 Jun 2021

Baochuan Lin, Ph.D. Academic Editor, PLOS ONE - PONE-D-21-05788R1

Dear Dr. Lin,

 We thank you and the reviewers for the positive and constructive comments on our manuscript. The individual comments and our responses are listed below.

Reviewer #1: Thanks for the opportunity to review the manuscript. The research idea was novel and attractive. However, the title is not completely compatible with the data and the methods that was provided in the manuscript. I have some comments and questions:

Answer: The title has been revised and now reads: Torquetenovirus in saliva: a potential biomarker for SARS-CoV-2 infection?

1- About the SARS-CoV-2 sampling, as I found the patients were instructed and did the sampling by themselves, using nasopharyngeal swabs. This method (nasopharyngeal swabs) is a sensitive sampling method but is painful and disgusting. How could you be sure that samples were collected correctly? I mean, how could you be sure that the negative samples were not false negative results?

Answer: As stated in the text, subjects were given a detailed video that clearly demonstrated the correct way to collect samples. Follow-up of subjects confirmed that all negative assays were truly negative. We included the term “nasal-oropharyngeal swab sample” in order to clarify the appropriated procedure which we used in this study. 

2- We are in the pandemic of COVID-19, could we assume a symptomatic individual as a control just with a single SARS-CoV-2 negative PCR?

Answer: As mentioned above we have had follow-up contact with all negative controls to verify that they were not infected. We also included a sentence in the lines 111 to 113 to clarify such aspect. 

3- It was a good idea to select the controls among symptomatic individuals who had negative SARS-CoV-2 PCR, but we need more information about the cause of disease in controls because an allergic rhinitis would be different from common cold. Moreover, some of the controls could have other strains of corona virus. I afraid the controls are not homogenous.

Answer: We thank the reviewer for this comment that requires clarification. The totality of symptoms in each of the controls were characteristic of the common cold. Those with allergic rhinitis were excluded. This is now stated in Methods (in the lines 85 to 86). We did not test for the presence of other coronavirus strains in the controls. Their presence was unlikely since the prevalence of these viruses is rare in our population.

4- In line 97, you wrote “Several individuals positive for SARS-CoV-2 in their saliva were requested to collect additional saliva samples over a 30-day period.”, what about individuals with negative PCR?

Answer: We only collected additional samples from SARS-CoV-2-positive patients. It probably would have been optimal to also collect additional samples from the controls. However, it was apparent that there was no clinical indication of symptom worsening or the presence of Covid in this group. 

5- About the clinical sign and symptoms, did the patient measured his/her body temperature? If yes, is this a sensitive method to divide patients to febrile and afebrile?

Answer: Body temperature was self-collected and, we agree, may not always be accurate. However, this was not the only parameter evaluated.

6- Could we have more information about the drug history and background disease of both cases and controls? Because it might affect the TTV viral load.

Answer: We have now added to the text in Methods that none of the subjects used antibiotics or other medications in the two weeks prior to sample collection (in the lines 87 and 88). In addition, they were all healthy with no underlying pathology. 

7- In table 3, you showed that the SARS-CoV-2-positive patients had the highest median TTV viral load on day 8-9 while the viral load was lower in day 6-7, how could you interpret this finding considering the normal immunologic response to SARS-CoV-2 infection?

Answer: The viral loads on days 6-7 and 8-9 are very similar and statistically indistinguishable.

8- Could you describe, why you divided the study course to days 3-5, 6-7, 8-9, 10-16,…? Why the intervals are not equal?

Answer: The interval was chosen to have approximately the same number of subjects at each time point. We believed that this provided the most reliable indication of changes.

9- You only provided data about the kinetics of SARS-CoV-2 virus, what about the clinical course of disease in different time points, data on hospital admission, or mortality?

Answer: As stated in the text, this is a study of patients with mild symptoms that resolved during the study period. None required hospitalization admission.

10- About the figure 1, this is a spaghetti plot showing the kinetics of TTV, if each shape is representative for a single patient, I afraid you had a lot of missing data. Could you provide the figure legend with more details about the plot and shapes?

Answer: The figure legend has now been implemented to state more explicitly that the graph measures the TTV titer in saliva at each time point for each subject where more than one sample was obtained (in the lines 213 to 216). 

11- You might need to provide the immunologic markers, as well as more information about the clinical course of disease, if you aim to conclude that TTV is a marker of local immune status and as predictors of prognosis.

Answer: Immune markers were not measured in these subjects and so we only speculate that the reduction in TTV as a predictor of prognosis occurred at least partially by a mechanism of immune activation. This was based on other studies relating TTV titer to immune system status. We have now clarified in Discussion that an immune-mediated mechanism was not directly tested and that our findings are consistent with an increased overall immune competence (in the lines 252 and 253). 

Reviewer #2: In this manuscript, Mendes-Correa et al. focuse on the attempt to demonstrate that TTV level in saliva from COVID-19 patients might be used as biomarker for progression or resolution of SARS CoV-2 infection. Although the topic may be of interest, the way it is treated is disappointing and a number of issues makes weak the manuscript. Overall, in the current form, the manuscript does not reach the quality standard needed for publication

Answer: We appreciate the reviewer’s concerns and trust that our responses to the specific comments pf the other reviewers will alleviate the reviewer’s concerns. 

Reviewer #3: Interesting study from a single center cohort about the association of TTV titer in saliva with SARS-CoV-2 status.

1) the major limitation is already discussed in the manuscript:

the SARS-CoV-2 negative group - all with "mild respiratory symptoms" was not further characterized.

Especially testing with a multiplex-PCR for other respiratory viruses would help to interpret the data in the manuscript, especially since TTV levels seem to stay unchanged in the SARS-CoV-negative group day 10-16 and >16 in contrast to the SARS-CoV-2 positive group.

If saliva samples are still available, I would recommend performing these analyses in both groups.

The manuscript could be further improved if a matched control group without respiratory symptoms would also be tested for TTV in saliva.

Answer: Unfortunately, the subjects in this study were not tested for other respiratory viruses and the samples are no longer available for this purpose. As the reviewer states, and as we acknowledged in Discussion, this as a study limitation. While it would certainly be of interest to determine the prevalence and titer of TTV in healthy individuals, and this is planned for the near future, that information is not essential for data analysis in the present investigation. 

2) line 49: it is true that TTV has been described as an indicator of immune function mainly after organ transplantation, but there is also literature on TTV as a marker of immune function in chronic diseases such as liver cirrhosis or HIV. This could be briefly elaborated by the authors to show that looking at TTV levels could also have a value outside of organ transplantation.

Answer: We thank the reviewer for this suggestion and have now added the additional information to Introduction with references (in the lines 55 to 57).

3) line 53/54: should be probably ref. 6 (TTV in HSCT recip.); "predicting lymphocyte status" is somewhat obscure to me, would suggest immune function, immunocompetence or immunreconstitution.

Answer: The text has been changed as suggested to now read “immunocompetence” and reference 8 has been added (in the line 61). 

4) line 191: it would be helpful to provide the number of subjects with more than one saliva sample, as this is rather small and therefore a cautious interpretation is required

Answer: The number of subjects with more than one saliva sample is evident from Figure 1. We have now added this, 10 subjects, to the text as suggested (in the lines 213 to 216).

5) line 203: would remove "anti-SARS-CoV-2" as no rationale is provided that TTV level reflects specific anti-SARS-CoV-2 immune activity - considering the literature, it is more likely a marker of overall immunocompetence

Answer: We agree, and has been stated in comment 11 for Reviewer #1, a mechanism revolving around altered immune status was speculative and not based on collection of immune-related parameters. The term “anti-SARS-CoV-2” has been deleted. 

Reviewer #4: The authors describe the evolution of torque tenovirus viral load in saliva related to , 10Sars-CoV 2 viral load in nasopharyngeal swabs, in an effort to find markers for immunosuppression. They also compare TTV levels at different times post initiation of symptoms for the covid 19 population (mild covid) and a control population with a mild respiratory disease.

Second, the authors show in a subpopulation of the covid 19 group that TTV viral load decreases with the disparition of symptoms. As a consequence, they consider that TTV viral load in saliva may be suitable to describe rhinopharyngeal local immune response. And they refer to studies comparing blood TTV viral load and evolution of immunity /rejection in transplant patients.

The idea is interesting, because simple tests to evaluate local immunity are very important to identify potential patients that could evolve toward high viral loads and covid worsening. The data concerning the evolution of TTV viral load in relation to SARS-CoV 2 viral load is convincing. This is, to my knowledge, the first paper describing TTV load in saliva for covid patients .

Answer: We thank the Reviewer for acknowledging our unique hypothesis regarding TTV analysis in Covid patients and its eventual clinical potential.

Although, the paper suffers from some weaknesses that could be overcome:

First : the authors did not justify the use of saliva instead of the Nasopharyngeal samples for TTV viral load. Although they refer to its use in previous comparisons with plasma levels in transplant patients to predict global immunity which is very different. They use the same extraction method . We understand of course that saliva is easier and potentially more reproducible for auto collection, but the TTV load values, in saliva, are highly variable in both populations and as NOP sampling was performed, they could have compared TTV load in both samples, to comfort their hypothesis.

Answer: We thank the reviewer for this comment. Unfortunately, the NOP samples were not available for TTV testing when the study was designed.

Second : The large panel of viral loads at each time, coming from the fact that there was only one sample per patient and thus at each time we compare different patients make the group comparison less significant, the authors should comment on this and provide a graph in addition to the table to show the repartition of viral loads within time in both groups.

Answer: We agree with the Reviewer that the use of saliva from different subjects at different time points is not optimal, and this is acknowledged in the Discussion as a study limitation as requested. We disagree with the Reviewer that in addition to the Table we should also provide a graph. We think this is unnecessarily repetitious and does not provide any additional information. 

Third : The lowering of TTV viral load from sequential samples in the covid patients cannot be compared with that of the non-covid group. In addition we do onot have the diagnosis (VRS may be different from rhinovirus for example in terms of local immunity, samely pharyngitis or thracheitis may be different from coryza?) Theu could add the diagnosis if available.

Answer: We responded to similar comments by Reviewer 1, comment #3 and Reviewer #3, comment 1 and agree that the control group might possibly be heterogeneous. We did not test the control subjects for the possible presence of different rhinoviruses as acknowledged as a study limitation in Discussion. Based solely on their symptoms and recovery time without treatment it appeared most likely that they had the common cold, and not an allergic rhinitis or a bacterial pharyngitis or tracheitis. We have now added a comment in Methods to further clarify this limitation. 

Fourth : The authors have choosen a mild covid group for a better comparison with mild respiratory disease. Could they correlate the initial TTV load or the maximum TTV load , with the duration of symptoms , or the delay to SARS-CoV 2 undetectability? This could be very interesting to predict persistent infections.

Answer: We thank the Reviewer for this very constructive suggestion. Unfortunately, in our study these additional comparisons were not possible. Our sample size was too small to perform this analysis with a high degree of validity. 

Minor comments : some text errors or form errors are present in the text and bibliography, and a cautious review is necessary before submission.

Answer: We apologize for these errors and have now corrected them in the revision. 

We are available to send you any further information you may require.

Yours faithfully,

Tania Regina Tozetto-Mendoza and co-authors.

---

## [Decision Letter · Decision Letter 1]

8 Jul 2021

PONE-D-21-05788R1

Torquetenovirus in saliva: a potential biomarker for SARS-CoV-2 infecton?

PLOS ONE

Dear Dr. Tozetto Mendoza,

Thank you for submitting your manuscript to PLOS ONE. After careful consideration, we feel that it has merit but does not fully meet PLOS ONE’s publication criteria as it currently stands. Therefore, we invite you to submit a revised version of the manuscript that addresses the points raised during the review process.

The revised manuscript has addressed most of the reviewers' comments, however, the reviewers still have concerns that need to be addressed.  Please see reviewers' comments below.

We look forward to receiving your revised manuscript.

Kind regards,

Baochuan Lin, Ph.D.

Academic Editor

PLOS ONE

Journal Requirements:

Reviewers' comments:

Reviewer's Responses to Questions

**Comments to the Author**

1. If the authors have adequately addressed your comments raised in a previous round of review and you feel that this manuscript is now acceptable for publication, you may indicate that here to bypass the “Comments to the Author” section, enter your conflict of interest statement in the “Confidential to Editor” section, and submit your "Accept" recommendation.

Reviewer #1: All comments have been addressed

Reviewer #3: All comments have been addressed

2. Is the manuscript technically sound, and do the data support the conclusions?

Reviewer #1: No

Reviewer #3: Yes

3. Has the statistical analysis been performed appropriately and rigorously? 

Reviewer #1: Yes

Reviewer #3: Yes

4. Have the authors made all data underlying the findings in their manuscript fully available?

Reviewer #1: No

Reviewer #3: Yes

5. Is the manuscript presented in an intelligible fashion and written in standard English?

Reviewer #1: Yes

Reviewer #3: Yes

6. Review Comments to the Author

Reviewer #1: Thank you for having addressed my comments. Although some of the responses could not satisfy me. For example about the response to comment 3, you responded, "We did not test for the presence of other coronavirus strains in the controls. Their

presence was unlikely since the prevalence of these viruses is rare in our population.". As I can see in literature, prevalence of human coronaviruses (HCoVs) is about 11.5% (https://doi.org/10.1590/S1517-83822013000100049).

Although the kinetics of TTV could be a possible marker of local immunity, but considering your results, it is difficult to predict prognosis using TTV.

Reviewer #3: While the lack of testing for other respiratory viruses and healthy controls somewhat limit the scientific value, the data should be still of interest for the scientific community.

7. PLOS authors have the option to publish the peer review history of their article (what does this mean?). If published, this will include your full peer review and any attached files.

Reviewer #1: **Yes: **Omid Rezahosseini

Reviewer #3: No

---

## [Author Response · Author response to Decision Letter 1]

17 Jul 2021

RESPONSE TO REVIEWERS

Baochuan Lin, Ph.D.

Academic Editor

PLOS ONE

Dear Dr. Lin:

 Thank you and the reviewers for the responses to our revised manuscript, PONE-D-21-05788R1, “Torquetenovirus in saliva: a potential biomarker for SARS-CoV-2 infection?” The remaining reviewer comments and our responses are listed below.

Reviewer #1

"About the response to comment 3, you responded, "We did not test for the presence of other coronavirus strains in the controls. Their presence was unlikely since the prevalence of these viruses is rare in our population". As I can see in literature, prevalence of human coronaviruses (HCoVs) is about 11.5% (https://doi.org/10.1590/S1517-83822013000100049)."

Answer: We appreciate the reviewer pointing out to us this relevant publication that we unfortunately had not previously seen. It is now cited in the revised text and included in the Reference section (reference 25, lines 341-343). The Discussion section has been extensively revised to more accurately reflect our observations and study limitations, as well as to bolster our argument that other viruses that may have been present in the nasopharynx likely did not have an appreciable influence on our findings. A portion of the revised Discussion is presented here: “Other viruses (reference 24), including different coronavirus species (reference 25), may also have been present in the nasopharynx of our study population. Thus, the possible influence of viral co-infections on SARS-C0V-2-related salivary TTV levels remains undetermined. We would stress, however, that the uniform responses in the SARS-CoV-2 positive and negative patients suggest that any influence of other co-infections is likely to be relatively minor. A parallel diminution in TTV titer over time after resolution of respiratory symptoms was not observed in the SARS-CoV-2-negative individuals. This strongly suggests that immune alterations associated with a mild respiratory tract infection are insufficient to have a noticeable influence on TTV levels in saliva”.

Reviewer #1: Although the kinetics of TTV could be a possible marker of local immunity, but considering your results, it is difficult to predict prognosis using TTV.

Answer: We agree with the reviewer that quantitation of TTV in saliva may not ultimately prove to be a sensitive indicator of SARS-CoV-2 prognosis. Our inclusion of a question mark at the end of the title of our manuscript signifies this possibility. It is our strong belief, however, that this possibility is worthy of further investigation and that publication of our observations will bring this possibility to the attention of the research community. We also believe that our revised Discussion further supports our argument that the kinetics of TTV is a valid maker of local immunity (in the manuscrip is in the lines 228 to 334 and lines 250 to 260). 

Reviewer #3

 While the lack of testing for other respiratory viruses and healthy controls somewhat limit the scientific value, the data should be still of interest for the scientific community.

Answer: We thank the reviewer for acknowledging the interest of our findings for the scientific community. Please see the comments to Reviewer #1 for our revisions that we believe further increase their scientific value.

We are available to send you any further information you may require.

Yours faithfully,

Tania Regina Tozetto-Mendoza and co-authors

---

## [Editor Report · Decision Letter 2]

22 Jul 2021

PONE-D-21-05788R2

Torquetenovirus in saliva: a potential biomarker for SARS-CoV-2 infecton?

PLOS ONE

Dear Dr. Tozetto Mendoza,

Thank you for submitting your manuscript to PLOS ONE. After careful consideration, we feel that it has merit but does not fully meet PLOS ONE’s publication criteria as it currently stands. Therefore, we invite you to submit a revised version of the manuscript that addresses the points raised during the review process.

The revised manuscript is scientifically sound, however, the quality of the language needs to be improved since PLoS ONE does not perform copyediting of manuscripts at any later stage in the publication process.<o:p></o:p>

We suggest you thoroughly copyedit your manuscript for language usage, spelling, and grammar. If you do not know anyone who can help you do this, you may wish to consider employing a professional scientific editing service.<o:p></o:p>

We look forward to receiving your revised manuscript.

Kind regards,

Baochuan Lin, Ph.D.

Academic Editor

PLOS ONE
---

## [Author Response · Author response to Decision Letter 2]

27 Jul 2021

Response to the editor comment

Baochuan Lin, Ph.D.

Academic Editor

PLOS ONE

Dear Dr. Lin:

Thank you for suggestion to the manuscript, PONE-D-21-05788R2, “Torquetenovirus in saliva: a potential biomarker for SARS-CoV-2 infection?” 

Our version of the manuscript PONE-D-21-05788R2 was reviewed by an experienced scientist whose native language is English, professor emeritus Steven S. Witkins. 

Yours faithfully,

Tania Regina Tozetto-Mendoza and co-authors

---

## [Editor Report · Decision Letter 3]

28 Jul 2021

PONE-D-21-05788R3

Torquetenovirus in saliva: a potential biomarker for SARS-CoV-2 infecton?

PLOS ONE

Dear Dr. Tozetto Mendoza,

Thank you for submitting your manuscript to PLOS ONE. After careful consideration, we feel that it has merit but does not fully meet PLOS ONE’s publication criteria as it currently stands. Therefore, we invite you to submit a revised version of the manuscript that addresses the points raised during the review process.

The revised manuscript has shown significant improvement, however, there quite a few things that still need edition/clarification.

Specific comments:

1. Line 37 & 38, suggest changing "more frequent" to "more prevalent".

2. Line 42, suggest changing "...TTV was initially present the TTV..." to "TTV was initially present, the TTV..."

3. Line 68, suggest adding "," after resolved.

4. Line 80, please use the same font.

5. Line 83 - 84, suggest deleting "Ethical approval was given to the project." Redundant statement.

6. Line 83, suggest changing "...informed written consent." to "...written informed consent."

7. Line 84, suggest changing "...a convenience sample..." to "...samples..."

8. Line 88 - 89, this sentence is awkward, please rephrase for clarity.

9. Line 90, suggest changing "They were all healthy with no underlying pathology." to "All subjects have no underlying condition."

10. Line 236, correct "All Individuals..." to "All individuals..."

11. Line 250, correct "An Alternate..." to "An alternate..."

We look forward to receiving your revised manuscript.

Kind regards,

Baochuan Lin, Ph.D.

Academic Editor

PLOS ONE
---

## [Author Response · Author response to Decision Letter 3]

30 Jul 2021

RESPONSE TO REVIEWERS - PONE-D-21-05788R4

Baochuan Lin, Ph.D.

Academic Editor

PLOS ONE

Dear Editor,

Thank you for the specific comments to our revised manuscript, PONE-D-21-05788R3, “Torquetenovirus in saliva: a potential biomarker for SARS-CoV-2 infection?” 

All changes were marked in blue color in the revised manuscript to each point raised by the academic editor. We apologize for these errors and have now corrected them in the revision.

Yours faithfully,

Tania Regina Tozetto-Mendoza, PhD

Laboratory of Virology,

Institute of Tropical Medicine

School of Medicine, University of São Paulo, Brazil

E-mail: tozetto@usp.br

July, 30, 2021

---

## [Editor Report · Decision Letter 4]

5 Aug 2021

Torquetenovirus in saliva: a potential biomarker for SARS-CoV-2 infecton?

PONE-D-21-05788R4

Dear Dr. Tozetto Mendoza,

We’re pleased to inform you that your manuscript has been judged scientifically suitable for publication and will be formally accepted for publication once it meets all outstanding technical requirements.

Kind regards,

Baochuan Lin, Ph.D.

Academic Editor

PLOS ONE
---

## [Editor Report · Acceptance letter]

16 Aug 2021

PONE-D-21-05788R4 

Torquetenovirus in saliva: a potential biomarker  for  SARS-CoV-2 infection? 

Dear Dr. Tozetto-Mendoza:

I'm pleased to inform you that your manuscript has been deemed suitable for publication in PLOS ONE. Congratulations! Your manuscript is now with our production department. 

Kind regards, 

on behalf of

Dr. Baochuan Lin 

Academic Editor

PLOS ONE